# Evaluation of Spliceosome Protein SmD2 as a Potential Target for Cancer Therapy

**DOI:** 10.3390/ijms252313131

**Published:** 2024-12-06

**Authors:** Jing Li, Peiyu Li, Tereza Brachtlova, Ida H. van der Meulen-Muileman, Henk Dekker, Vishal S. Kumar, Marieke Fransen, Idris Bahce, Emanuela Felley-Bosco, Victor W. van Beusechem

**Affiliations:** 1Amsterdam UMC location Vrije Universiteit Amsterdam, Medical Oncology, De Boelelaan 1117, 1081 HV Amsterdam, The Netherlands; 2Cancer Center Amsterdam, Cancer Biology and Immunology, 1081 HV Amsterdam, The Netherlands; 3ORCA Therapeutics BV, Onderwijsboulevard 225, 5223 DE ‘s-Hertogenbosch, The Netherlands; 4Amsterdam Institute for Infection and Immunity, Cancer Immunology, 1081 HV Amsterdam, The Netherlands; 5Amsterdam UMC location Vrije Universiteit Amsterdam, Pulmonary Medicine, De Boelelaan 1117, 1081 HV Amsterdam, The Netherlands; 6Cancer Center Amsterdam, Cancer Treatment and Quality of Life, 1081 HV Amsterdam, The Netherlands; 7Department of Biomedical Sciences, University of Lausanne, Rue du Bugnon 7, CH-1005 Lausanne, Switzerland

**Keywords:** Sm proteins, pan-cancer analysis, *SNRPD2* dependency, drug sensitivity, mitosis

## Abstract

The core spliceosome Sm proteins are gaining attention as potential targets for cancer treatment. Here, we evaluate this, with focus on SmD2. A pan-cancer analysis including 26 solid tumor types revealed that the SmD2-encoding *SNRPD2* gene was overexpressed in almost all cancers. In several cancers, high *SNRPD2* expression was associated with a poor prognosis. To investigate the vulnerability of human cells to the loss of SmD2 expression, we silenced *SNRPD2* using a short hairpin-expressing lentiviral vector in established cancer cell lines; in short-term cultured melanoma cells; and in several normal cell cultures, including cancer-associated fibroblasts cultured from non-small cell lung cancer resections. Additionally, we analyzed publicly available cell viability datasets for the dependency of cancer cell lines to SmD2 expression. Together, these studies clearly established SmD2 as a cancer-selective lethal target. Delving into genes with similar essentiality profiles to *SNRPD2*, we uncovered the intersected lethal stress between the loss of SmD2 and the loss of gene products participating in not only different mRNA processing steps including mRNA splicing, but also processes for coordinated protein production, as well as mitosis. Furthermore, we could correlate *SNRPD2* expression to the responses of cancer cells to several FDA-approved anti-tumor drugs, especially to drugs inhibiting the cell cycle. Overall, our study confirms the anticipated role for targeting SmD2 in cancer treatment and reveals non-canonical SmD2 functions beyond mRNA splicing that could contribute to the dependency of cancer cells to high *SNRPD2* expression.

## 1. Introduction

RNA splicing, as a fundamental cellular process, plays a pivotal role in the regulation of gene expression by orchestrating the removal of intronic regions from pre-mRNA and the precise joining of exons to form mature mRNA [1]. The spliceosome machinery that recognizes specific sequences at the boundaries between introns and exons and catalyzes the chemical RNA splicing reactions consists of a variety of small nuclear ribonucleoproteins (snRNPs) and other protein components [2]. The dysregulation of RNA splicing in cancer leads to the generation of aberrant mRNA isoforms, contributing to the diversity of the cancer transcriptome. This emerges as a critical player in cancer biology, influencing the molecular mechanisms underpinning tumorigenesis, progression, and drug resistance [3,4,5,6]. Previously, we have reviewed spliceosome mutations and alterations in splicing patterns in lung cancer [7]. These alterations can affect critical genes involved in cell cycle regulation, apoptosis, and other cellular processes, contributing to the development and progression of cancer. In this context, exploring the complex connections between RNA splicing dysregulation and cancer not only sheds light on the molecular intricacies of oncogenesis but also gives opportunities for the development of targeted therapeutic strategies.

The core structure of Sm in the spliceosome consists of seven Sm proteins (SmB/B’, SmD1, SmD2, SmD3, SmE, SmF, and SmG) that form a ring-like structure around small nuclear RNA (snRNA) [8]. This ring serves as a scaffold for the assembly of snRNP particles, facilitating the formation of the active spliceosome. Previously, we have shown that the inhibition of any of the seven Sm proteins using RNA interference (RNAi) effectively kills non-small cell lung cancer (NSCLC) cells but not non-malignant lung cells [9]. Furthermore, the silencing of *SNRPD3* (encoding SmD3) in an NSCLC cell line, but not in lung fibroblasts, produced distinct alternative splicing switches in the ADD3 gene from a variant that is associated with cancer progression to a variant predominantly found in non-malignant tissue [9]. In addition, the silencing of any of the Sm genes induced a cytotoxic splicing switch in the proteasomal subunit beta type-3 (*PSMB3*) gene in NSCLC cells, but not in non-malignant lung cells [10]. Hence, Sm proteins, essential components of the snRNPs in the spliceosome, have gained attention as potential targets in lung cancer therapy.

In this study, using SmD2 as an example, we investigate the therapeutic effect of targeting Sm genes in cancer. We show that *SNRPD2* is overexpressed in almost all cancers and that this is associated with poor prognoses in several cancers. To expand our analysis of vulnerability to the loss of SmD2 beyond lung cancer, we include cells from many other cancer types as well as several different non-malignant cell cultures originating from healthy tissue and from tumor stroma. Our studies demonstrate broad cancer-selective vulnerability to the loss of *SNRPD2* expression. In addition, we present an in silico analysis of the sensitivity of a large panel of human cancer cell lines to the knockdown or knockout of *SNRPD2*; of genes and biological processes that have *SNRPD2*-like effects on the viability of these cancer cells; and of *SNRPD2* expression-related susceptibility to anti-tumor drugs.

## 2. Results

### 2.1. SNRPD2 Is Overexpressed in Most Solid Tumors and Is a Potential Prognostic Biomarker Across Multiple Cancer Types

We compared Sm gene expression at the mRNA level between tumors and their matched normal tissues from 26 solid tumor types by using The Cancer Genome Atlas (TCGA) Program (https://www.cancer.gov/tcga (accessed on 30 June 2024)) and Genotype-Tissue Expression Project (GTEx) datasets [11]. As can be seen in Figure 1a, significantly higher *SNRPD2* expressions were observed in all cancers, except for a rare kidney cancer type, chromophobe renal cell carcinoma. These findings paralleled with those for the other six Sm genes (Appendix A). Survival analysis revealed poor overall survival (OS) and progression-free intervals (PFIs) among patients with high *SNRPD2*-expressing adrenocortical carcinoma, clear cell and chromophobe cell renal carcinomas, low-grade glioma, lung adenocarcinoma, mesothelioma and uveal melanoma; and shorter PFIs among patients with high *SNRPD2*-expressing renal papillary cell carcinoma and hepatocellular carcinoma (Figure 1b, Appendix A). In particular, in uveal melanoma, high *SNRPD2* expression appeared to be a very strong risk factor. In contrast, SmD2 appeared to be a modest protective factor prolonging OS among breast invasive carcinoma and lung squamous cell carcinoma patients (Figure 1b, Appendix A).

The apparent opposite prognostic value of *SNRPD2* expression in lung adenocarcinoma (LUAD) versus lung squamous cell carcinoma (LUSC), both subtypes of NSCLC, was striking (Figure 1b). To discover potential biological factors that cause the difference, we conducted hallmark gene set enrichment analysis (GSEA) [12,13] on the *SNRPD2* gene among LUAD and LUSC tissues. In this analysis, genes with expressions correlating to that of *SNRPD2* were ranked and compiled in hallmark gene sets (Appendix A). Hallmarks that correlated with SmD2 expression in both LUAD and LUSC included, among others, MYC and E2F target genes, oxidative phosphorylation genes, G2M checkpoint genes, DNA repair genes, and mTORC1 signaling genes (Figure 1c, left). In contrast, UV and inflammatory response gene sets exhibited the most prominent negative correlation with *SNRPD2* expression (Figure 1c, left). Since these gene sets correlated similarly with *SNRPD2* expression in LUAD and LUSC, their associated pathways and processes are not likely responsible for the contrasting correlation between *SNRPD2* expression and clinical response in these NSCLC types. The most prominent difference between the GSEA results on LUSC and LUAD was that LUSC exhibited a strong negative correlation between *SNRPD2* expression and epithelial–mesenchymal transition (EMT) and interferon (IFN) response gene sets (Figure 1c, right). Thus, in contrast to LUAD, LUSC with high *SNRPD2* expression exhibited low EMT and IFN responses. This could perhaps provide clues as to why high *SNRPD2* expression seems to be a protective marker in LUSC but not in LUAD, but the correlation could also be coincidental.

### 2.2. Dependency of Human Cancer Cell Lines on SNRPD2 Expression

Previously, we demonstrated cancer-selective cytotoxicity of Sm gene silencing on NSCLC cell lines [9]. The overexpression of *SNRPD2* in almost all cancers and the association of this overexpression with poor prognoses in several cancers suggest that the lethal effect of Sm gene silencing might not be specific to NSCLC cells but a more general effect that could potentially also be used to treat other types of cancer. To investigate this, we transduced a panel of 25 human cancer cell lines of various tissue origins with a lentiviral vector (LV) expressing a short hairpin targeting *SNRPD2* (LV-shSNRPD2) or with an empty control LV (LV-EV) (Figure 2a). The panel included eight lung cancer cell lines, six of which were tested with siRNA silencing *SNRPD2* before, and 17 cell lines representing cancers originating from the skin, pancreas, esophagus, colon, breast, and prostate. Six days after transduction, cell viabilities were measured using CellTiter-Blue. As can be seen in Figure 2a, most cancer cell lines tested were sensitive to silencing *SNRPD2*. This indicates that the inhibition of SmD2 may have wider utility to treat not only NSCLC but also other cancers.

To find further evidence for the presumed dependency of cancer cells on SmD2 expression for survival, we analyzed the publicly available Cancer Dependency Map dataset that compiles results of RNAi and CRISPR loss-of-function screens for cancer cell viability (https://depmap.org/portal/ (accessed on 21 February 2024)) [14,15]. The collected data from RNAi screens included 346 human cancer cell lines from a variety of tissue origins, including 113 lung cancers. Sensitivity to *SNRPD2* knockdown was variable (Figure 2b, Appendix A). The median score for SmD2 on all cell lines (−0.97) corresponded to the median of all essential genes (which is by default set to −1). RNAi screen data for the other six Sm genes (*SNRPB*, *SNRPD1*, *SNRPD3*, *SNRPE*, *SNRPF*, and *SNRPG*) showed similar sensitivities to knockdown, with median dependency scores ranging from −1.66 to −0.72 (Appendix A). The Sm genes are thus considered essential genes in human cancer cells. The dependency of NSCLC cells on Sm gene expression was similar to that of the complete cancer cell line dataset (Appendix A). Hence, the known cytotoxicity of Sm gene silencing on NSCLC cells was representative for the general vulnerability of cancer cells to this interference. Sensitivity to *SNRPD2* knockdown by RNAi exhibited a weak positive correlation with *SNRPD2* expression (Pearson correlation analysis, *r* = 0.23, *p* < 0.0001; Figure 2c). This suggests that cancer cells upregulate *SNRPD2* expression because they depend on it and that this makes them more vulnerable to the inhibition of expression. In the CRISPR screen dataset, 469 human cancer cell lines were included, among which were 119 lung cancer cell lines. As expected for an essential gene, all these cancer cell lines were found to be strongly dependent on an intact *SNRPD2* gene (median Chronos dependency score = −2.10, i.e., considerably lower than for RNAi screens; Figure 2b). In line with the notion that CRISPR screens test the effect of complete gene knockout, cell line dependencies did not correlate with *SNRPD2* expression levels (Pearson correlation analysis, *p* = 0.32; Figure 2d).

### 2.3. Silencing SNRPD2 Kills Primary Short-Term Cultured Melanoma Cells

Primary cancer cells more closely mimic the physiological diversity and genetic heterogeneity of the tumors from which they are derived than established cancer cell lines [16,17]. Therefore, we examined the killing effect of silencing *SNRPD2* in patient-derived metastatic melanoma cell cultures. For these studies, we established three cultures from different patients; Mel 40a cells were cultured from a skin metastasis, Mel 41 cells from an abdominal lymph node metastasis, and Mel 131 cells from axillary lymph node metastases. The cell cultures were characterized using FACS for melanoma antigen recognized by T-cells (MART1) expression and (myo)fibroblast marker alpha smooth muscle actin (α-SMA) (Figure 3a). While cultured Mel 40a and Mel 131 cells expressed MART1 and were negative for α-SMA as is expected for melanoma cells, Mel 41 cells did not express the melanoma marker MART1, but approximately 20% of the cells were positive for α-SMA. From the Mel 41 lymph node suspensions, we thus most probably cultured a heterogenic mixture of cells including α-SMA-positive lymph node fibroblastic reticulum cells. It is unknown if the 80% MART1 and α-SMA double-negative cell population represents MART1-negative metastatic melanoma cells or non-fibroid stromal cells. We consider the latter more likely, because MART1-negative melanoma metastases are rare [18]. To test the effect of *SNRPD2* silencing on the three cell cultures, they were transduced with LV-EV or LV-shSNRPD2 at 4000 genome copies (gc)/cell (Figure 3b,c). *SNRPD2* knockdown (60–70%) was confirmed 3 days after lentiviral transduction by quantitative reverse transcription PCR (RT-qPCR) (Figure 3b). One week after transduction, cell viability was measured using CellTiter-Blue. As can be seen in Figure 3c, silencing *SNRPD2* effectively killed the two primary melanoma cell cultures Mel 40a and Mel131. In contrast, Mel 41 cells that were presumably not melanoma cells were insensitive to *SNRPD2* silencing. For Mel 40a cells, we also monitored cell proliferation and apoptosis over a period of 7 days using live cell imaging (Figure 3d,e). Untransduced and LV-EV transduced primary melanoma cells remained alive and proliferated to reach near-confluence over the observation period, whereas LV-shSNRPD2 transduced cells exhibited apoptosis induction and severely inhibited population growth. Hence, heterogeneous primary melanoma cultures, but not stroma cells isolated from a melanoma-infiltrated lymph node, also experience the cytotoxic effect of Sm gene silencing.

### 2.4. Silencing SNRPD2 Does Not Reduce Viability of Non-Malignant Cells

To more elaborately investigate the impact of *SNRPD2* silencing on the viability of non-malignant cells, we examined a larger panel of human cell cultures. This included three non-malignant cell lines from diverse tissue origins—i.e., Human Umbilical Vein Endothelial Cells (HUVECs), Human Pancreatic Stellate Cells (HPaSteCs), and Human Fetal Lung fibroblasts (HFL1)—and five patient-derived lung fibroblast cultures, four of which were grown from NSCLC resections. To characterize the patient-derived cells, we used flow cytometry and immunocytochemistry to assess the expression of α-SMA and Pan-Cytokeratin (Appendix A). This revealed that the cell cultures consisted mainly of fibroblasts and were almost or completely devoid of carcinoma cells. In all non-malignant cell cultures except HUVECs, endogenous *SNRPD2* expression was approximately 3-fold lower than that measured in the melanoma cultures (Appendix A). Pilot experiments using an LV-expressing enhanced Green Fluorescent Protein (LV-eGFP) showed that an LV dose of 4000 gc/cell efficiently transduced most of the cells in all the tested cell cultures (Appendix A). Therefore, all the cell cultures were transduced with LV-EV or LV-shSNRPD2 at 4000 gc/cell and analyzed using the same methods as were used for melanoma cell cultures (Figure 4). Three days after transduction, the knockdown of *SNRPD2* was confirmed, and after one week, cell viability was measured. On different cell cultures, a 50–80% reduction in *SNRPD2* expression was observed (Figure 4a). None of the cell cultures exhibited a detectable decrease in cell viability (Figure 4b). Hence, in contrast to cancer cells, all the tested non-malignant cells survived *SNRPD2* gene silencing.

### 2.5. Biological Processes and Functionally Related Genes with Essentiality Profiles Similar to SNRPD2

It has been suggested that a similar essentiality profile among different genes in loss-of-function screens indicates that they share common functional units or pathways [19]. Therefore, in search of an explanation for the vulnerability of cancer cells for the loss of SmD2, we performed gene ontology enrichment analysis for biological processes that exhibit codependency with SmD2 in CRISPR screens included in the Cancer Dependency Map dataset. As expected, this showed the highest enrichment values for mRNA splicing processes, followed by protein production and degradation and other mRNA processing steps (Figure 5, Appendix A). To find functionally related genes with essentiality profiles similar to SmD2 in RNAi screens, we used the ShinyDepMap web tool [19]. RNAi connectivity analysis revealed 39 genes clustering with SmD2 (listed in Appendix A). The gene ontology enrichment analysis for biological processes, in which 35 of these functionally related genes are involved (4 genes not covered in DAVID), highlighted only processes associated with the identified proteasome and ribosome genes (Appendix A).

### 2.6. Association Between SNRPD2 Expression and Anti-Tumor Drug Sensitivity

Given the non-canonical functions that SmD2 intersected with, we next investigated if its expression level predicts the utility of currently available anti-tumor drug applications by using public resources from the Genomics of Drug Sensitivity in Cancer (GDSC) database (https://www.cancerrxgene.org (accessed on 11 April 2024)) [20]. The area under the dose–response curve (AUC) values of FDA-approved anti-tumor drugs (including targeted drugs and chemotherapies, Appendix A) and the genomic data from 969 cancer cell lines were collected for gene expression comparisons and constructing Receiver Operating Characteristic (ROC) curve plots (Figure 6 and Appendix A). Cell lines within the lower tertile were considered as sensitive responders, and those in the upper tertile were considered as resistant non-responders [21]. For several drugs, areas under the ROC curve > 0.6 were found that suggest a possible helpful risk discrimination [22]. For comparison, ROC curves were generated on the same dataset for the known drug–target interactions of sorafenib with PDGFRA, PDGFRB, PDGFRL, RAF1, BRAF, KIT, and RET (Appendix A). This presented AUC values below 0.6. As can be seen in Figure 6, cancer cells responding to treatment with the EGFR inhibitor gefitinib had lower *SNRPD2* expression than non-responders (AUC = 0.67, ROC *p* < 0.0001). For cells treated with other EGFR inhibitors (erlotinib and osimertinib) or with MEK inhibitors (trametinib and selumetinib), a similar association between *SNRPD2* expression and sensitivity was observed, albeit with a lower AUC (AUC = 0.58–0.59, ROC *p* < 0.001; Appendix A). This suggested that one outcome of high *SNRPD2* expression in cancer cells could be downstream activation of EGFR/MEK pathways causing resistance to EGFR and MEK inhibitors. In contrast, the opposite association of high *SNRPD2* expression with sensitivity to treatment was found for the multiple receptor tyrosine kinase inhibitors sorafenib (targeting PDGFR, KIT, VEGFR, and RAF; AUC = 0.61, ROC *p* < 0.0001) and crizotinib (targeting MET, ALK, and ROS1; AUC = 0.60, ROC *p* < 0.0001), the mitosis inhibitor alisertib targeting AURKA (AUC = 0.62, ROC *p* < 0.0001), the HDAC inhibitor vorinostat (AUC = 0.68; ROC *p* < 0.0001), the microtubule inhibitors vinorelbine, vinblastine, and vincristine (AUC = 0.63–0.64; ROC *p* < 0.0001), the platinum-based alkylating agents oxaliplatin and cisplatin (AUC = 0.61–0.62; ROC *p* < 0.0001), the anti-tumor antibiotics dactinomycin and mitoxantrone (AUC = 0.61–0.62; ROC *p* < 0.0001), the topoisomerase inhibitor irinotecan (AUC = 0.61; ROC *p* < 0.0001), and the proteasome inhibitor bortezomib (AUC = 0.59, ROC *p* < 0.001; Appendix A). These findings are consistent with the results of the gene ontology enrichment analysis showing that SmD2 is functionally linked to mitosis, protein production, and protein degradation. They suggest that high SmD2 expression promotes these processes, making cancer cells more vulnerable to their pharmacological inhibition.

To investigate if the observed associations are specific to SmD2 or more common for Sm proteins, ROC analysis was also performed for the other Sm genes in the dataset, i.e., all except *SNRPE*. Figure 7 presents the anti-tumor drugs that exhibited an AUC > 0.6 with ROC *p* < 0.0001. As can be seen, high or low expression of many Sm genes was a classifier for sensitivity to anti-cancer drugs, with AUC values ranging between 0.6 and 0.8. While sensitivity to some drugs (i.e., bleomycin, fulvestrant, talazoparib, and methotrexate) correlated with the expression of only a single Sm gene, for many drugs a significant correlation was found with multiple Sm genes. Most strikingly, sensitivity to gefitinib, selumetinib, and trametinib was associated with low expressions of almost all Sm genes; whereas the vulnerability of cancer cells to dactinomycin, vinblastine, vincristine, irinotecan, and vorinostat was highly reproducible if they had higher expressions of any of the Sm genes (Figure 7). Hence, many observed associations were common, suggesting canonical rather than non-canonical roles for the Sm proteins in their sensitivity to these anti-cancer drugs.

## 3. Discussion

Investigating the dysregulation of the spliceosome in cancer provides valuable insights into cancer biology and might reveal new footholds for therapy. Previous high-throughput RNAi screens carried out in our laboratory discovered several components of the spliceosome as putative targets for non-small cell lung cancer treatment. In particular, we found that Sm gene expression was associated with NSCLC grade and stage and that NSCLC cells were more vulnerable to silencing Sm genes than were non-malignant lung cells [9]. Interestingly, recent investigations revealed that Sm genes are overexpressed in a multitude of cancers [23,24,25,26,27] and that the depletion of SmD2, one of the spliceosome core components that we prioritized for targeted treatment development, inhibited the proliferation of hepatocellular carcinoma cells [24] and triple-negative breast cancer cells [28].

In this study, we performed an in silico pan-cancer analysis of *SNRPD2* expression in patient samples, comparing gene expression to prognosis and to hallmarks of cancer. This demonstrated that *SNRPD2* was generally overexpressed in tumors, was a putative prognostic biomarker in several cancers, and was associated with the activation of multiple gene sets that are considered hallmarks of oncogenesis and cancer progression. We also investigated the effects of *SNRPD2* knockdown on a panel of cancer cell lines derived from different tumor tissues and on primary short-term cultured melanoma cells and non-malignant cells (vascular endothelial cells, pancreatic stellate cells, lymph node stromal cells, and lung fibroblasts, including cancer-associated fibroblasts from lung cancer resections). In addition, we explored the dependency of a large panel of cancer cell lines on *SNRPD2* expression by performing in silico analysis of public databases that collect cell viability data from gene knockdown and knockout screens. Across the board, we observed that cancer cells—in contrast to non-malignant cells—are highly vulnerable to the loss of *SNRPD2* expression. This confirmed that targeting SmD2 has cancer-selective therapeutic utility, extending our previous observations on lung cancer cells to multiple solid tumor types and to a larger collection of clinically relevant cell cultures.

As a structural component of the core spliceosome, SmD2 should be considered essential for cell survival. The complete loss of SmD2 is expectedly lethal for any cell. The observation that the silencing of *SNRPD2* or one of the other Sm genes—in contrast to most other spliceosome genes—is selectively lethal to cancer cells [9] was thus surprising. Here, we found that non-malignant cells could cope with very low *SNRPD2* expression levels. They were not harmed by a reduction in *SNRPD2* expression to levels up to at least 15-fold less than the expression measured in untreated cancer cells. Conversely, cancer cells did not survive an approximately 3-fold reduction in *SNRPD2* expression, i.e., to levels similar to those in untreated non-malignant cells. Hence, lethality was not caused by a reduction in *SNRPD2* expression below a critical level needed for cell survival. Instead, vulnerability to *SNRPD2* depletion appeared synthetically lethal with malignancy. This is in line with a proposed non-oncogene addiction of cancer cells, which allows them to maintain dysregulated mRNA splicing programs via increased spliceosome activity to cope with cancer-associated cellular stresses [7].

To obtain some insight in the biological processes that are associated with the vulnerability of cancer cells to the loss of *SNRPD2* expression, we performed a gene enrichment analysis for functionally related genes with similar essentiality profiles in cancer cells as *SNRPD2*. This presented not only the mRNA splicing process but also other mRNA processing steps; protein translation, stability, and degradation; and mitosis. Following the notion that similar essentiality profiles reveal functional relationships [19], this suggests that SmD2 may have functional properties in other cellular processes besides pre-mRNA splicing. There is a growing body of evidence supporting this idea. In eukaryotic cells, Sm proteins were not only found to be associated with snRNAs and with small Cajal body RNAs that guide post-transcriptional modification of snRNAs but also with mature intronless mRNAs [29]. It was suggested that Sm proteins facilitate the transport of bound mRNAs to specific microdomains in the cytoplasm [30] and that this sorting of mRNAs allows cells to temporarily store highly expressed transcripts for later translation [31]. Interestingly, an analysis of mRNAs associated with Sm proteins identified two major categories, one of which was transcripts encoding ribosome and translation-related proteins [29]. It is thus tempting to speculate that the depletion of SmD2 dysregulated the cytoplasmic sorting of mRNAs encoding proteins of the translation machinery needed for their coordinated production and thus had similar lethal effects on cancer cells as knocking out the genes encoding these proteins. Also, the significant codependency of SmD2 with mitotic centrosome separation that we observed fits with previous observations. In a siRNA library screen for human cell division, the knockdown of two Sm genes, albeit not *SNRPD2*, delayed mitosis [32], thus pointing at a role for Sm proteins in facilitating progression through mitosis. The effect of Sm proteins on mitosis seems indirect, via alternative splicing of the transcript encoding an essential component of the sister chromatid cohesion complex [28,33]. Silencing several Sm genes including *SNRPD2* weakened the association of cohesin with chromatin, causing chromosome misalignment and faulty segregation during mitosis [28,33].

Interestingly, in our analysis correlating *SNRPD2* expression to anti-tumor drug sensitivity on a large panel of human cancer cell lines, we observed that cancer cells that were sensitive to treatment with several FDA-approved targeted drugs expressed higher *SNRPD2* levels than resistant cancer cells. This was the case for an AURKA inhibitor that regulates mitotic centrosome maturation and chromosome segregation; an HDAC inhibitor that remodels chromatin structure; three microtubule inhibitors that block mitosis; two platinum-based alkylating agents; and two anti-tumor antibiotics and a topoisomerase inhibitor that interfere with DNA replication and transcription. These observations are in line with a putative non-canonical function of SmD2 in promoting cell division or with SmD2 driving alternative splicing of transcripts encoding proteins involved in mitosis. Our observation that similar correlations existed for the other Sm genes suggests that the latter is the more likely explanation. A higher cell proliferation rate could make high Sm-protein-expressing cancer cells more vulnerable to antimitotic treatments. A similar response trend was observed for the treatment of cancer cells with the proteasome inhibitor bortezomib. This was consistent with the findings in our gene ontology analysis revealing codependency between SmD2 and protein degradation processes. Hence, high SmD2 expression possibly increases proteasome activity, making cells more vulnerable to proteasome inhibition. This would be in line with our previous finding that the silencing of Sm genes induced a cancer-selective lethal splicing switch in the *PSMB3* gene encoding one of the proteasome proteins [10]. This splicing switch reduced the expression of the functional full-length PSMB3 protein, which in turn was associated with a decrease in proteasome activity. Thus, silencing Sm genes caused a cytotoxic effect similar to that of bortezomib treatment, with the crucial difference that it was more cancer-specific due to the alternative splicing event occurring only in cancer cells [10].

An opposite association between *SNRPD2* expression and susceptibility to treatment was observed for EGFR and MEK inhibitors, with resistant cells exhibiting high *SNRPD2* expression. While mechanistically explaining this observation is beyond the scope of our present work, we hypothesize that SmD2 perhaps activates the signaling pathways downstream of EGFR and MEK, making the inhibition of these targets ineffective. Previously, the silencing of one of the other Sm proteins (SmD1) was shown to suppress the PI3K/Akt/mTOR pathway in hepatocellular carcinoma cells [34]. Together, these observations suggest that Sm proteins promote cancer by activating signaling pathways that drive cell proliferation. They also suggest that it is worthwhile to investigate if combined Sm protein targeting with EGFR and/or MEK inhibition could have synergistic anti-tumor effects.

## 4. Materials and Methods

### 4.1. Pan-Cancer Datasets and Analyses

The genomic expression data (RNAseq-Toil RESM TPM) and clinical phenotype data from the TCGA Pan-Cancer dataset and the Sm gene expression data from the GTEx dataset were downloaded from the UCSC Xena platform (UCSC Xena (xenabrowser.net)) on 30 June 2024 [35]. The data we used were standardized and with batch effects removed [36]. Tumor types and matched normal tissues with sample sizes greater than 3 were included for Sm gene expression level comparisons. Cancer types included (with their abbreviations according to TCGA) were as follows: ACC: adrenocortical carcinoma; BLCA: bladder urothelial carcinoma; BRCA: breast invasive carcinoma; CESC: cervical squamous cell carcinoma and endocervical adenocarcinoma; CHOL: cholangiocarcinoma; COAD: colon adenocarcinoma; ESCA: esophageal carcinoma; GBM: glioblastoma multiforme; HNSC: head and neck squamous cell carcinoma; KICH: kidney chromophobe; KIRC: kidney renal clear cell carcinoma; KIRP: kidney renal papillary cell carcinoma; LGG: brain lower-grade glioma; LIHC: liver hepatocellular carcinoma; LUAD: lung adenocarcinoma; LUSC: lung squamous cell carcinoma; MESO: mesothelioma; OV: ovarian serous cystadenocarcinoma; PAAD: pancreatic adenocarcinoma; PRAD: prostate adenocarcinoma; READ: rectum adenocarcinoma; SKCM: skin cutaneous melanoma; STAD: stomach adenocarcinoma; TGCT: testicular germ cell tumors; THCA: thyroid carcinoma; UCEC: uterine corpus endometrial carcinoma; UCS: uterine carcinosarcoma. The *SNRPD2* expression in patients with different cancer types retrieved from the TCGA Pan-Cancer dataset was found to be correlated to clinical survival (OS and PFI) by using the Cox Proportional Hazards Model. Gene expression in cancer specimens was found to be correlated to *SNRPD2* expression using Spearman’s correlation; and the output gene list was used for hallmark GSEA by using the hallmarks gene set “h.all.v2023.2.Hs.entrez.gmt” from the Molecular Signatures Database [12,37]. The Pan-Cancer data were processed and plotted using R packages (data.table, plotly, dplyr, ggpubr, survival, ggrepel and ggplot2 (CRAN: Contributed Packages); clusterProfiler and org.Hs.eg.db (Bioconductor—3.20 AnnotationData Packages)) and R software (version 4.4.1).

### 4.2. Cell Lines and Culture Conditions

A549, SW1573, (NCI-)H292, (NCI-)H1299, (NCI-)H1650, (NCI-)H460, (NCI-)H1703, and (NCI-)H2228 NSCLC cell lines, BRO and SKMEL28 melanoma cell lines, PANC-1, HPAC, and BXPC3 pancreas carcinoma cell lines, OE33, OE19, and OE21 esophagus carcinoma cell lines, HCT116, COLO205, and HT29 colon carcinoma cell lines, SKOV-3 ovarian cancer cell line, MDA-MB-231 and MCF7 breast cancer cell lines, PC3, 22Rv1, and DU145 prostate cancer cell lines, HPaSteC pancreatic stellate cells, HUVEC umbilical vein endothelial cells, and HEK293T cells were obtained from the cell line repository of the Laboratory Medical Oncology, Amsterdam UMC, The Netherlands. HFL1 fetal lung fibroblasts were obtained from the Department of Molecular Cell Biology and Immunology, Amsterdam UMC, The Netherlands. All cell lines were tested negative for mycoplasma every 3 months. A549, SW1573, H460, HT29, MDA-MB-231, MCF7, 22Rv1, HPaSteC, and HEK293T cell lines were maintained in Dulbecco’s Modified Eagle’s Medium high glucose (Sigma-Aldrich Chemie NV, Zwijndrecht, The Netherlands) and all other cancer cell lines were maintained in RPMI 1640 (Sigma-Aldrich Chemie NV, Zwijndrecht, The Netherlands), supplemented with 10% Fetal Bovine Serum (FBS; Gibco, Fisher Scientific, Landsmeer, The Netherlands) and 1% penicillin/streptomycin (P/S; Sigma-Aldrich, St. Louis, MO, USA). HFL1 cells were maintained in Ham’s F-12K (Kaighn’s) Medium (Gibco, Fisher Scientific, Geel, Belgium) supplemented with 10% Newborn Calf Serum (NBCS; Capricorn Scientific, Ebsdorfergrund, Germany) and 1% P/S. HUVECs were maintained in Medium 199 (M199; Sigma-Aldrich Chemie GmbH, Darmstadt, Germany) supplemented with 10% NBCS and 2% human serum (a kind gift of Judy van Beijnum, Laboratory Medical Oncology, Amsterdam UMC, Amsterdam, The Netherlands), 1% GlutaMAX™ Supplement (Gibco, Thermo Fisher Scientific, Bleiswijk, The Netherlands), and 1% Antibiotic-Antimycotic (A/A; Gibco, Fisher Scientific, Landsmeer, The Netherlands). During experiments, P/S was omitted from the medium. All culturing procedures were performed at 37 °C with 5% CO_2_.

### 4.3. Generation of Patient-Derived Melanoma Cell Cultures and Culture Conditions

Patient-derived melanoma cell cultures were established from long-term cryopreserved cell suspensions made from melanoma metastases resected from three advanced-stage patients who participated in a clinical study of autologous whole-cell vaccination at the VU University medical center between 1987 and 1998 [38]. Mel 40a cells were derived from a metastatic skin lesion; Mel 41 cells were derived from an abdominal lymph node metastasis; and Mel 131 cells were derived from axillary lymph node metastases (right + left). To establish cell cultures, 2 × 10^5^ cells were cultured in RPMI-1640 medium supplemented with 10% fetal calf serum (Hyclone Laboratories, Logan, UT, USA), 100 IU/mL penicillin, 100 µg/mL streptomycin, 2 mM L-glutamine (P/S/Glut, Gibco, Fisher Scientific, Landsmeer, The Netherlands), and 0.05 mmol/L 2-β-mercaptoethanol (2-ME, Merck, Kenilworth, NJ, USA) per well in a 48-well plate (Greiner Bio-One, Alphen aan den Rijn, The Netherlands). Attached cells were expanded with media refreshment every 2–3 days. Once the cell cultures reached 80% confluence, they were seeded in larger flasks for further expansion. After approximately two months of expansion, fully established patient-derived cell cultures were examined for the expression of MART1 and α-SMA (Section 4.6). During experiments, P/S/Glut and 2-ME were omitted from the medium. All culturing procedures were performed at 37 °C with 5% CO_2_.

### 4.4. Generation of Patient-Derived Lung Fibroblast and Cancer-Associated Fibroblast Cell Cultures

Patient-derived lung fibroblasts were cultured from resections of patients with (suspected) lung cancer who underwent surgery at the Amsterdam UMC, location VUmc. PDLF1 cells were derived from a primary lung lesion resection specimen. Histological and molecular analysis revealed that the resection material was a solitary fibrous tumor. Patient-derived lung cancer-associated fibroblast cultures 1 to 4 were derived from primary lesion resection specimens from patients with NSCLC. PDLCAF1 cells were from a patient with stage IIIA squamous cell carcinoma; PDLCAF2 cells were from a patient with stage I adenocarcinoma; PDLCAF3 cells were from a patient with stage I mucinous adenocarcinoma; and PDLCAF4 cells were from a patient with stage I adenosquamous carcinoma.

Resections were placed in a 50 mL sterile conical tube containing DMEMglx (DMEM, high glucose, GlutaMAX™ Supplement, pyruvate, Gibco, Thermo Fisher Scientific, Bleiswijk, The Netherlands) with 10% Bovine Serum Albumin (BSA, Merck Life Science NV, Amsterdam, The Netherlands) and 1% P/S on wet ice during transportation from the operating room to the research laboratory. Upon arrival, the resections were minced manually using a sterile scalpel and a tweezer, and this was followed by a digestion step at 37 °C for 1 h using the pre-set program ‘37C_h_TDK_2’ on the gentleMACS™ Octo Dissociator with Heaters, with Miltenyi Biotec’s Human Tumor Dissociation Kit (Miltenyi Biotec BV, Bergisch Gladbach, Germany). Dissociation was stopped by adding 5 mL of complete primary cell culture medium consisting of DMEM/F12 (HEPES-buffered Dulbecco’s Modified Eagle Medium/Nutrient Mixture F-12, Gibco, Thermo Fisher Scientific, Bleiswijk, The Netherlands) supplemented with 1% Insulin-Transferrin-Selenium-Ethanolamine (ITS-X; Gibco, Thermo Fisher Scientific, Bleiswijk, The Netherlands), 1% GlutaMAX™ Supplement (Gibco, Thermo Fisher Scientific, Bleiswijk, The Netherlands), 10 ng/mL Epidermal Growth Factor (EGF; Corning, Amsterdam, The Netherlands), 10 ng/mL Cholera Toxin (Merck Life Science NV, Amsterdam, The Netherlands), 100 nM Hydrocortisone (Merck Life Science NV, Amsterdam, The Netherlands), 0.1 nM Triiodothyronine (Merck Life Science NV, Amsterdam, The Netherlands), 0.5% BSA, 5% FBS, and 1% Antibiotic-Antimycotic (Gibco, Fisher Scientific, Landsmeer, The Netherlands). Subsequently, a 5 min red blood cell lysis step was performed using red blood cell lysis buffer (Roche Diagnostics Deutschland GmbH, Mannheim, Germany). After digestion and lysis, cell pellets were resuspended in complete primary cell culture medium, and cell suspensions were seeded in a 48-well plate (SARSTEDT, Nümbrecht, Germany). After 7–10 days, when the attached cells reached 80–90% confluence, they were harvested and expanded in DMEM/F12 supplemented with 1% ITS-X, 1% GlutaMAX™, and 10% NBCS. Established patient-derived cell cultures were characterized by immunocytochemistry (Section 4.5) and flow cytometry (Section 4.6). All culturing procedures were performed at 37 °C with 5% CO_2_.

### 4.5. Immunocytochemistry Assay

The cells were seeded in a 96-well cell culture plate (SARSTEDT, Nümbrecht, Germany), 2000 cells/well. The next day, the cells were washed with PBS and fixed with 4% paraformaldehyde (PFA, SERVA Electrophoresis GmbH, Heidelberg, Germany) for 20 min at room temperature (RT) and permeabilized with 0.5% Triton X-100 (Merck Life Science NV, Amsterdam, The Netherlands) in PBS for 5–10 min at RT, and this was followed by three washing steps with PBS and a 1 h blocking step with PBS containing 2% BSA. After another three washes with PBS, the primary antibody against α-SMA (clone 1A4, Agilent Technologies DAKO, Glostrup, Denmark) or Pan-Cytokeratin (clone AE1/AE3 + 5D3, Abcam Netherlands BV, Amsterdam, The Netherlands) diluted 1:500 in PBS with 2% BSA was added and incubated for 20–24 h at 4 °C. Thereafter, a mouse-specific HRP/DAB (ABC) Detection IHC Kit (Abcam Netherlands BV, Amsterdam, The Netherlands) was used. The cells were incubated in biotinylated goat anti-mouse antibody for 10–20 min at RT, washed 3 times in PBS, and stained with DAB Chromogen and DAB Substrate for 5 min, and these were followed by three final washing steps 3 with PBS. Pictures were captured under a Nikon Ti2 microscope using NIS-Elements imaging software 1.21.00 and processed using ImageJ 1.50i.

### 4.6. Flow Cytometry

The cells harvested by trypsinization were fixed with 1% PFA for 30 min at RT and permeabilized with 0.1% (*w*/*v*) saponin (Acros Organics, Fisher Scientific, Geel, Belgium) in PBS for 20 min at RT. After fixation and permeabilization, primary antibodies against α-SMA diluted 1:50 in PBS containing 0.1% BSA and Pan-Cytokeratin diluted 1:100 in PBS containing 0.1% BSA or Melan-A/MART-1 (MLANA/788, Novus Biologicals, Abingdon, UK) diluted 1:100 in PBS containing 0.1% BSA was added and incubated for 1 h at 4 °C. Melan-A/MART-1 is FITC-labeled, and an FITC-labeled mouse IgG1 kappa isotype control (P3.6.2.8.1, eBioscience, Life Technologies, Bleiswijk, The Netherlands) serves as the negative control. For α-SMA and Pan-Cytokeratin, APC-labeled goat anti-mouse IgG secondary antibody (Poly4053, BioLegend Europe BV, Amsterdam, The Netherlands) diluted 1:500 in PBS containing 0.1% BSA was applied for 30 min at 4 °C. Here, the negative control was cells stained with secondary antibody only. Thereafter, flow cytometry was performed using an FACSCalibur (BD Biosciences, San Jose, CA, USA), and acquired events were analyzed with FlowJo software 10.8.1.

### 4.7. Lentiviral Vector Production and Cell Transduction

Lentiviral vectors LV-EV and LV-shSNRPD2 were made by transfecting 3 × 10^6^ HEK293T cells with negative control transfer vector pLKO.1 (Thermo Fisher Scientific Open Biosystems, Landsmeer, The Netherlands, #RHS4080) or lentiviral transfer vector #TRCN0000074400 expressing an shRNA silencing *SNRPD2* (guide strand sequence 5’-CATCAACTGCCGCAACAATAA-3′; Thermo Fisher Scientific Open Biosystems, Landsmeer, The Netherlands), respectively, together with pMD2.G (a gift from Didier Trono; Addgene, Watertown, MA, USA, #12260) and psPAX2 (a gift from Didier Trono; Addgene, Watertown, MA, USA, #12259) packaging constructs (4 μg mix of 1778 ng psPAX2, 222 ng pMD2.G, and 2000 ng transfer vector), using FuGENE^®^ HD (Promega Benelux, Leiden, The Netherlands) transfection reagent at 3 μL per μg DNA in serum-free OptiMEM medium (Gibco, Thermo Fisher Scientific, Landsmeer, The Netherlands). The next day, the culture medium was changed to DMEM with 30% FBS and 1% P/S. Two days after transfection, the culture medium containing virus particles was harvested and cleared via centrifugation at 1250 rpm for 5 min at RT. Enhanced Green Fluorescent Protein-expressing lentiviral vector LV-eGFP was made using transfer vector pLenti6.2eGFP (a kind gift of Bart Westerman, Department of Neurosurgery, Amsterdam UMC, Amsterdam, The Netherlands) and procedures similar to the above, transfecting a 10 μg mix of 3750 ng psPAX2, 1250 ng pMD2.G, and 5000 ng transfer vector into 2 × 10^6^ HEK293T cells. Virus was harvested after 2 and 3 days in DMEM/F12 supplemented with 30% NBCS and 1% P/S.

LV preparations were concentrated using PEG-it™ Virus Precipitation Solution (System Bioscience SBI, Sanbio B.V., Uden, The Netherlands). The functional transduction unit (TU) titer of LV-EV was determined based on the capacity to confer puromycin resistance to SW1573 cells. Lentivirus genome copy titers were determined using the Lenti-X™ qRT-PCR Titration Kit (Takara Bio Europe SAS, Bio-Connect B.V., Huissen, The Netherlands) according to the manufacturer’s recommendations.

Cancer cell lines were plated at 1000 cells per well in 96-well plates and (depending on the susceptibility of the cell line to lentiviral vector transduction) subjected to 300 or 600 TU LV-EV or LV-shSNRPD2 per cell in culture medium with 4 μg/mL polybrene (Sigma-Aldrich, St. Louis, MO, USA). Non-malignant cell lines, patient-derived melanoma cells, patient-derived lung fibroblast cells, and patient-derived lung cancer-associated fibroblast cells were plated at 5000 cells per well in 96-well plates and subjected to 4000 genome copies of LV-EV or LV-shSNRPD2 per cell in culture medium with 6 μg/mL polybrene. One day after transfection, the medium was replaced with fresh medium, and the cells were cultured until analysis.

### 4.8. Cell Viability Assay

The cells were seeded and transduced the next day with LV-EV or LV-shSNRPD2 as described above. On the sixth or seventh day after transduction as indicated, 20 µL of CellTiter-Blue (Promega, Madison, WI, USA) reagent was added, and the cells were incubated for 3 h. The reaction was terminated by adding 50 µL 3% sodium dodecyl sulfate in deionized water, and cell viability was determined by measuring fluorescence at 540 nm excitation and 590 nm emission wavelengths using a Tecan Infinite F200 reader (Tecan Group, Männedorf, Switzerland).

### 4.9. IncuCyte Live Cell Imaging Assay

Mel 40a cells were seeded and transduced the next day with LV-EV or LV-shSNRPD2 as described above. One day after transduction, the culture medium was replaced with RPMI-1640 supplemented with 30% FBS and 5 µM IncuCyte^®^ Caspase-3/7 Green Dye (Essen Bioscience, Welwyn Garden City, UK). The cells were imaged in an IncuCyte^®^ ZOOM Live-Cell Analysis System (Essen Bioscience) with a 10× objective over 7 days, with pictures taken every 4 h. The images were analyzed using IncuCyte ZOOM software (v2018A; Essen Bioscience, Redwood Shores, CA, USA) according to the manufacturer’s protocol.

### 4.10. Quantitative Reverse Transcription Polymerase Chain Reaction Analysis

Cell pellets from knockdown experiments were collected 3 days after lentiviral vector transduction. RNA was prepared using the RNeasy Plus Kit (Qiagen, Hilden, Germany) according to the manufacturer’s protocol. Total RNA was reverse-transcribed using the FIREScript RT cDNA synthesis KIT (Solis Biodyne, Bio-Connect, Huissen, The Netherlands). Real-time PCR was performed on the Roche LightCycler^®^ 480 System (Roche Diagnostics Nederland B.V., Almere, The Netherlands) using the HOT FIREPol^®^ EvaGreen^®^ qPCR Mix Plus (no ROX) (Solis BioDyne, Bio-Connect, Huissen, The Netherlands) according to the manufacturer’s protocol. Primers were designed using Primer-BLAST 3.0 (RRID:SCR_003095) and purchased as custom oligonucleotides from Invitrogen (*ACTB*: forward primer: 5′-TTCCTATGTGGGCGACGAG-3′, reverse primer: 5′-TCCTCGGGAGCCACACG-3′; *SNRPD2*: forward primer: 5′-ATGAGCCTCCTCAACAAGCC-3′, reverse primer: 5′-GTGAGCACAGAGAGTGGACC-3′). Relative mRNA levels compared to *ACTB* household gene expression were determined on the basis of the threshold cycle calculated by LightCycler 480 software using the ∆Ct method.

### 4.11. Analysis of DepMap SmD2 RNAi and CRISPR Loss-of-Function Screening Datasets

The dataset of loss-of-function screens using RNAi and CRISPR/Cas9 libraries provided by the DepMap Achilles project [14,15] was analyzed. Estimated knockdown effect scores from RNAi screens (Achilles+DRIVE+Marcotte, DEMETER2), knockout effect scores from CRISPR screens (DepMap 23Q4 Public+Score, Chronos), and expression levels (DepMap Public 23Q4) of individual genes in respective cancer cell lines were downloaded from the DepMap portal (https://depmap.org/portal/depmap/ (accessed on 21 February 2024)). Functionally related genes with essentiality profiles similar to that of SmD2 RNAi were clustered using the ShinyDepMap web tool (settings: small cluster size, probability threshold 0.9 (accessed on 21 February 2024)) [19]. Gene ontology enrichment analysis was performed using the DAVID online platform (https://david.ncifcrf.gov (accessed on 21 February 2024)) [39,40].

### 4.12. Analysis of Anti-Tumor Drug Responses Correlated to SmD2 Expression

The drug sensitivity and gene expression data in different cancer cell lines were obtained from the GDSC database (https://www.cancerrxgene.org (accessed on 11 April 2024)) [20]. The FDA-approved anti-tumor drugs and their AUC values from CellTiterGlo assays on cancer cell lines were obtained from the GDSC2 database (Release 8.5) on 11 April 2024, and the Sm expression data were retrieved from the RMA normalized Affymetrix Human Genome array downloaded from the GDSC1000 resource on 11 April 2024. Box plots and ROC curves were used for visualizing drug responses correlated to Sm gene expression, and these were processed by R packages (dplyr, ggpubr, plotly, ggplot2, cutoff (cutoff points were calculated using “Youden index = Sensitivity + Specificity − 1”), and pROC) and R software (version 4.4.1).

### 4.13. Statistical Analyses

All data were analyzed using GraphPad Prism software version 9.3.1 (471) or R software version 4.4.1. The statistical tests applied are given with the description of the results.

## 5. Conclusions

We found that *SNRPD2* is overexpressed in most human solid tumors and that silencing it is lethal for cancer cells of many tissue origins including short-term cultured patient-derived melanoma cells, but not for non-malignant cells including cancer-associated fibroblasts. Together, this confirmed our expectation that SmD2 is a promising selective lethal therapeutic target for wide application in cancer. The identification of biological processes and pathways on which cells are similarly dependent as they are on the expression of *SNRPD2* provided clues to begin understanding the cancer-selective lethality of silencing Sm genes. Our findings suggest that the dependency of cancer cells on high *SNRPD2* expression might be explained by the dysregulation of mitosis and protein turnover in cancer cells, either via alternative splicing or through non-canonical functions of Sm proteins. This notion was further supported by correlations found between SNRPD2 expression and sensitivity to several anti-cancer drugs targeting mitosis and protein degradation. A more complete understanding of the vulnerability of cancer cells to the loss of Sm protein functions requires further research.

## Figures and Tables

**Figure 1 ijms-25-13131-f001:**
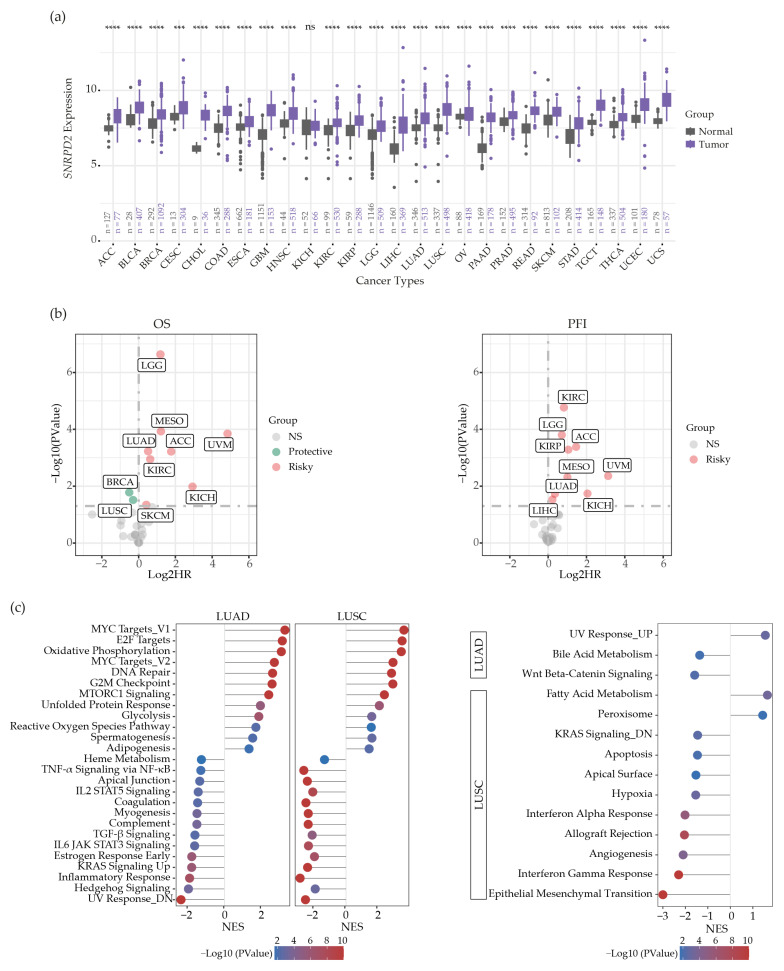
A pan-cancer analysis of *SNRPD2* gene expression. (**a**) *SNRPD2* mRNA expression comparison among 26 solid cancer types and their matched normal tissues. The gene expression data were retrieved from TCGA Pan-Cancer and GTEx datasets (unit: log2(TPM+0.001)). The sample size for each group in each comparison is marked by “n”. Differences between normal tissues and tumor tissues were tested by unpaired Student’s *t*-test (ns, not significant; ***, *p* < 0.001; ****, *p* < 0.0001). (**b**) Analysis of potential prognostic value of *SNRPD2* expression. The clinical phenotype data were retrieved from TCGA Pan-Cancer datasets. The Cox regression model was used to calculate hazard ratios (HRs) of *SNRPD2* expression for overall survival (OS; left panel) and progression-free interval (PFI; right panel). Grey dot: effect of *SNRPD2* expression is not significant; green dot: *SNRPD2* is a protective factor (*p* < 0.05 and HR < 1); pink dot: *SNRPD2* is a risk factor (*p* < 0.05 and HR > 1). (**c**) *SNRPD2* hallmark GSEA on LUAD and LUSC tissues. Hallmark gene sets that are commonly associated with *SNRPD2* expression in both LUAD and LUSC are plotted in lollipop format (left panel); the hallmark gene sets specific to LUAD or LUSC are presented in lollipop format (right panel). NES: normalized enrichment score. Cancer type abbreviations used (see Section 4.1) are according to TCGA.

**Figure 2 ijms-25-13131-f002:**
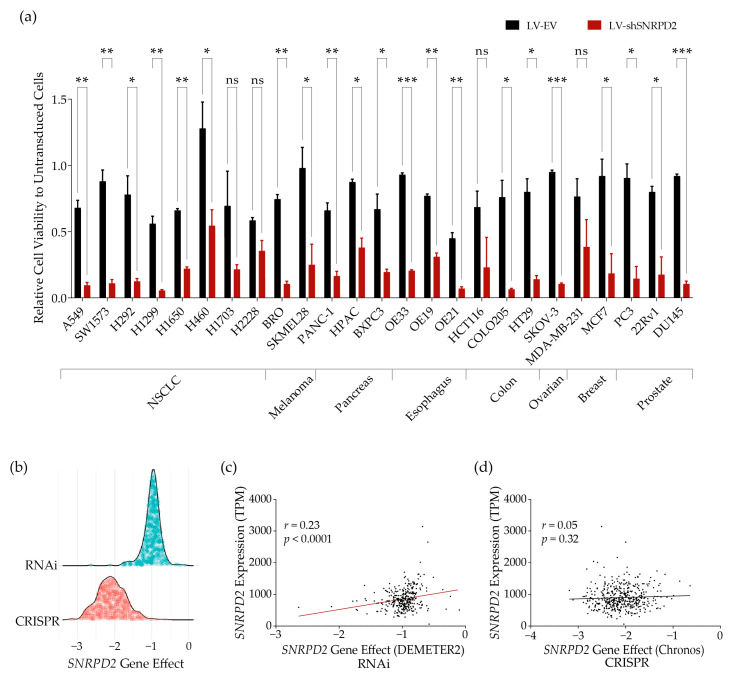
*SNRPD2* dependency analysis across multiple cancer cell lines in vitro. (**a**) Susceptibility of different cancer cell lines to the inhibition of *SNRPD2* gene expression. A panel of human cancer cell lines was transduced with an LV silencing *SNRPD2* or with a negative-control LV. For every cell line, viabilities were normalized by the matched untransduced control. The data shown are the means + SD of two independent experiments performed in triplicate. Differences between viabilities of cells transduced with LV-EV and LV-shSNRPD2 were tested by unpaired Student’s *t*-test (ns, not significant; *, *p* < 0.05; **, *p* < 0.01; ***, *p* < 0.001). (**b**) *SNRPD2*’s effect on cancer cell viability from RNAi and CRISPR loss-of-function screens. RNAi and CRISPR dependency scores of *SNRPD2* were retrieved from the DepMap RNAi (Achilles+DRIVE+Marcotte, DEMETER2) dataset and the CRISPR (DepMap 23Q4 Public+Score, Chronos) dataset. A score of 0 is equivalent to a gene that is not essential; a score of -1 corresponds to the median of all common essential genes. The lowest score indicates the most sensitive cells. (**c**) Correlation between *SNRPD2* expression levels in untreated controls and sensitivity to *SNRPD2* RNAi (Pearson correlation analysis, *p* < 0.0001; dots represent the different cancer cell lines; the red line shows best fit correlation). (**d**) Correlation between *SNRPD2* expression levels in untreated controls and sensitivity to *SNRPD2* gene knockout (Pearson correlation analysis, *p* > 0.05; dots represent the different cancer cell lines; the black line shows best fit correlation).

**Figure 3 ijms-25-13131-f003:**
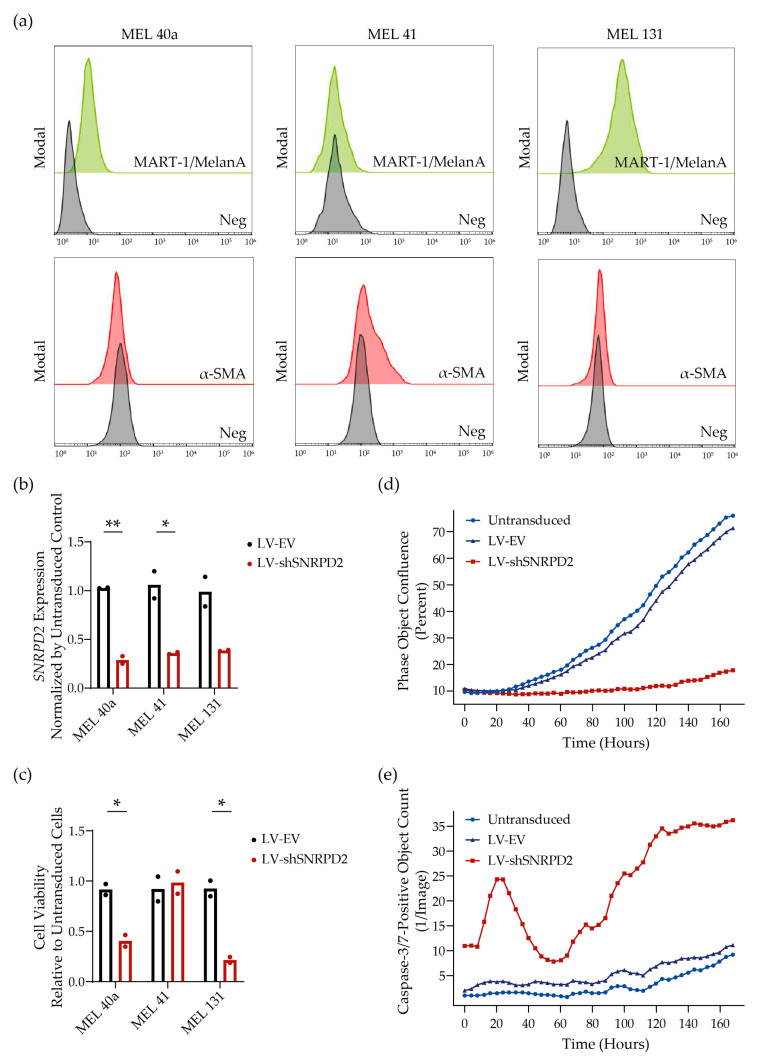
Lethal effect of *SNRPD2* gene silencing on primary short-term cultured melanoma cells. (**a**) Characterization of three short-term cultures of cells isolated from melanoma metastases, by flow cytometry analysis for MART1 and α-SMA expression. (**b**,**c**) *SNRPD2* expression (**b**) and cell viability (**c**) in three short-term cultures of cells isolated from melanoma metastases treated with LV-shSNRPD2 or control LV-EV. Data shown are from two independent experiments conducted in triplicate and are shown relative to untransduced control cells. *SNRPD2* expression was analyzed by RT-qPCR and normalized by *ACTB* expression three days after transduction; cell viability was assessed one week after LV transduction using CellTiter-Blue. Differences between LV-EV and LV-shSNRPD2 treated cells were tested by unpaired Student’s *t*-test (*, *p* <0.05; **, *p* < 0.01). (**d**,**e**) Live cell imaging of Mel 40a cultures using IncuCyte S3 Live Cell Analysis System. Cells were monitored for 7 days after transduction with LV-shSNRPD2 or control LV-EV. (**d**) Cell proliferation monitored by measuring monolayer confluency. (**e**) Apoptosis monitored using Caspase-3/7 Green Dye fluorescence.

**Figure 4 ijms-25-13131-f004:**
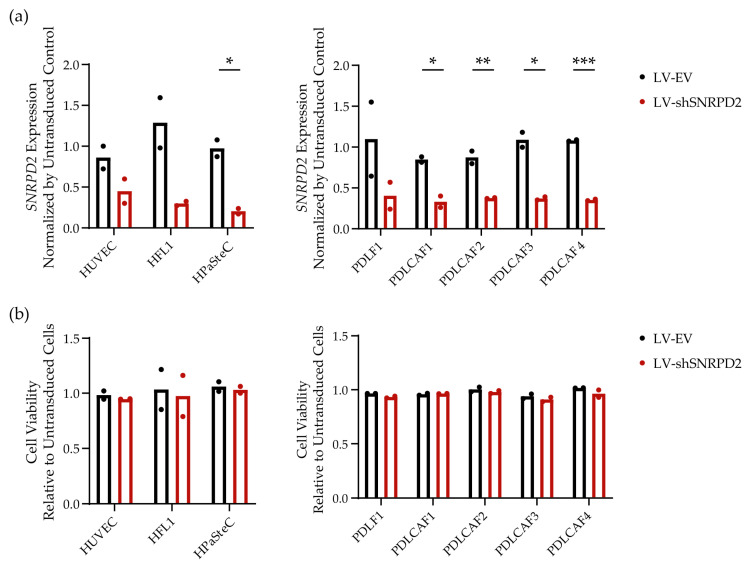
Non-lethality of *SNRPD2* gene silencing on non-malignant cells. (**a**) *SNRPD2* expression in non-malignant cell cultures transduced by LV-shSNRPD2 or control LV-EV. *SNRPD2* expression was analyzed by RT-qPCR and normalized by *ACTB* expression three days after lentiviral transduction. (**b**) Cell viability assay on non-malignant cell cultures one week after transduction with LV-shSNRPD2 or control LV-EV. All data shown are from two independent experiments conducted in triplicate and are normalized by the results of untransduced control cells. Differences between LV-EV-treated and LV-shSNRPD2-treated cells were tested by unpaired Student’s *t*-test (*, *p* < 0.05; **, *p* < 0.01; ***, *p* < 0.001).

**Figure 5 ijms-25-13131-f005:**
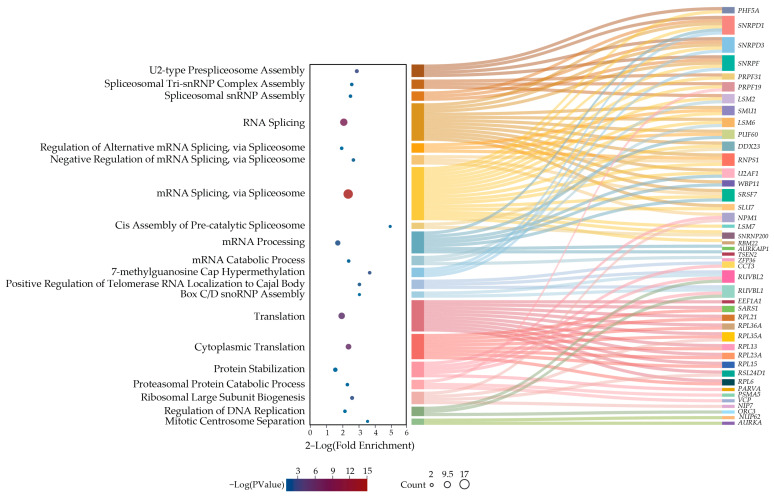
Gene ontology enrichment analysis of codependent biological processes with *SNRPD2* knockout in the CRISPR DepMap portal. The figure includes all GO terms with *p* < 0.05. They are connected to the enriched genes categorized under each term. The GO terms can be classified as part of one of four main biological processes: mRNA splicing (marked brown-yellow), RNA processing (marked blue), protein turnover (marked red-pink), and cell cycle (marked green). The enriched genes are indicated by rectangular blocks with different colors.

**Figure 6 ijms-25-13131-f006:**
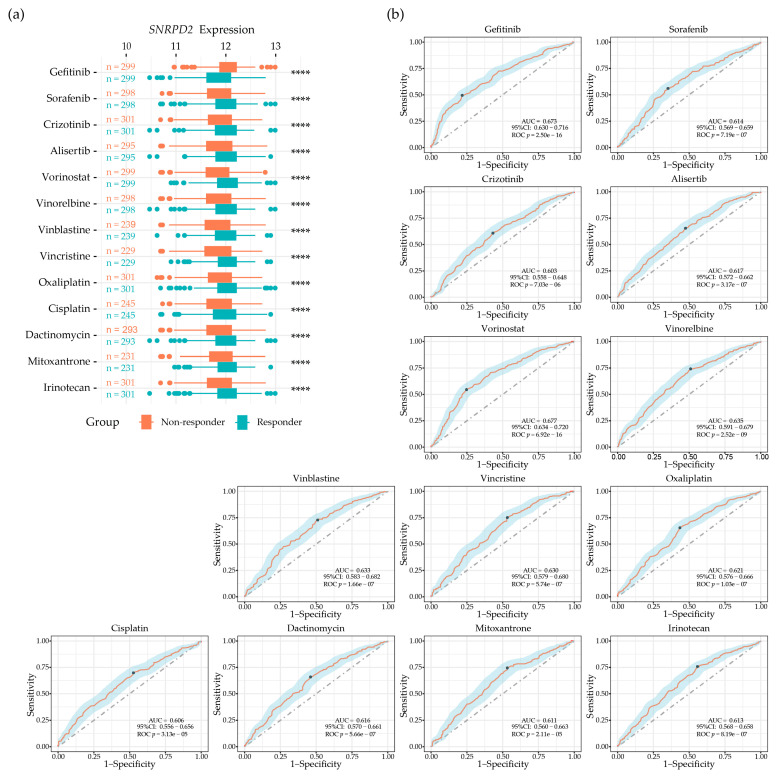
*SNRPD2* expression predicts the response of cancer cells to treatment with anti-tumor drugs in vitro. Box plots (**a**) and ROC curves (**b**) of *SNRPD2* expression and response to FDA-approved anti-tumor drugs. The *SNRPD2* expression data (RMA normalized Affymetrix Human Genome array) for cancer cell lines were downloaded from the GDSC1000 resource. A CellTiterGlo assay was used for cell viability and AUC evaluation, and AUC values were extracted from the GDSC2 database. *SNRPD2* expressions in non-responder and responder groups were compared using the Mann–Whitney U test (**a**) and ROC test (**b**). In (**a**): ****, *p* < 0.0001; in (**b**): the strongest cutoff is shown as a black dot on the ROC curves. The 95% confidence interval for the AUC is shown as a light blue ribbon. The figure includes drugs for which a significant association between *SNRPD2* expression and sensitivity with AUC < 0.6 was observed. Drugs with notable associations that did not meet this criterion are shown in Appendix A. The results for all drugs analyzed are given in Appendix A.

**Figure 7 ijms-25-13131-f007:**
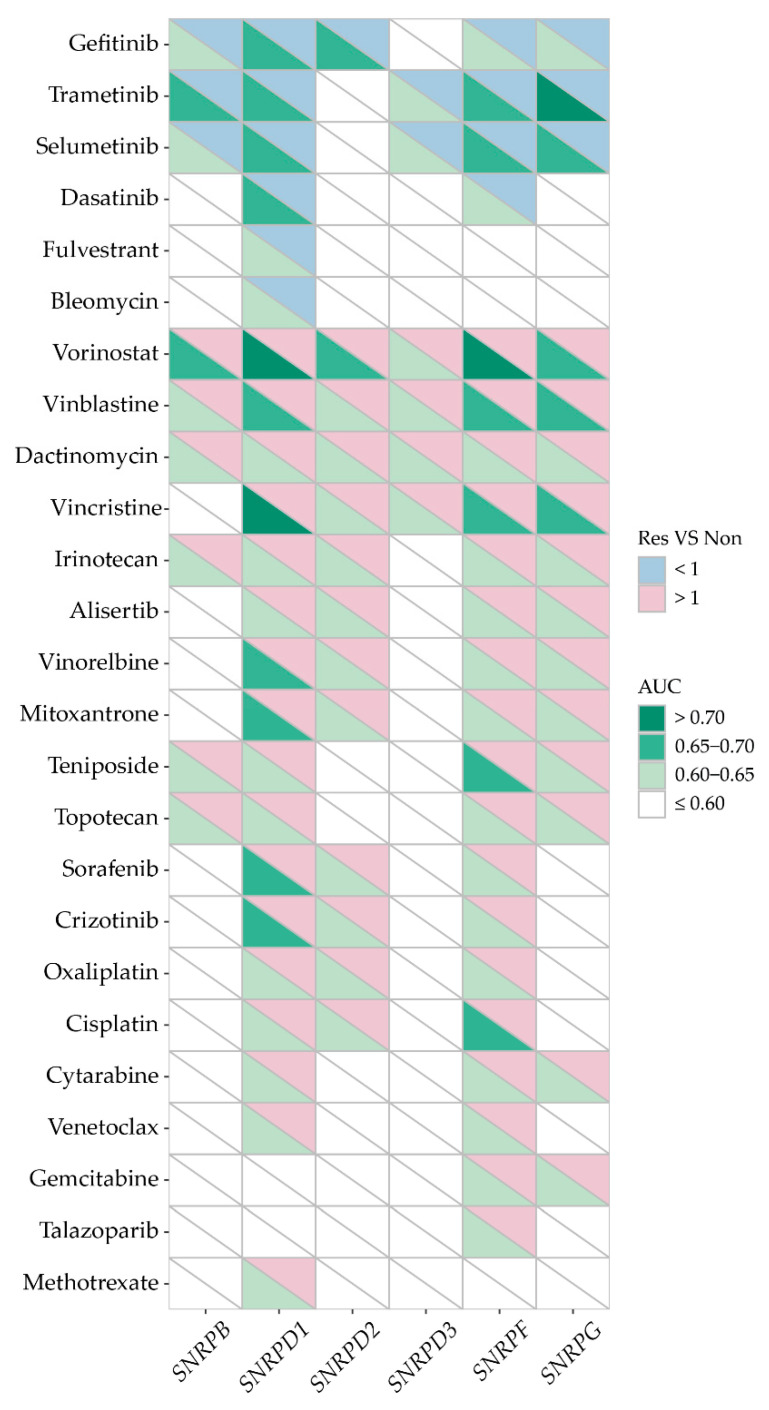
Comparison of Sm gene (*SNRPB*, *SNRPD1*, *SNRPD2*, *SNRPD3*, *SNRPF*, and *SNRPG*) expressions and responses to anti-tumor drugs in cancer cells. The Sm gene expression data (RMA normalized Affymetrix Human Genome array) for cancer cell lines were downloaded from the GDSC1000 resource. A CellTiterGlo assay was used for cell viability and AUC evaluation, and AUC values were extracted from the GDSC2 database. Sm gene expressions in non-responder and responder groups were compared using the Mann–Whitney U test and ROC test. The figure includes drugs for which a significant association between Sm gene expression and sensitivity with an AUC > 0.6 (Mann–Whitney U test *p* < 0.0001 and ROC *p* < 0.0001) was observed. The diagonal heatmap was plotted using the ratio of mean Sm gene expression in responders over non-responders (Res VS Non; upper triangle) and the AUC (lower triangle).

## Data Availability

The data presented in this study are available in this article and Appendix A.

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
