# Peer review of "Evaluation of Spliceosome Protein SmD2 as a Potential Target for Cancer Therapy"

_ijms, 2024, doi:10.3390/ijms252313131_

Round 1

Reviewer 1 Report

Comments and Suggestions for Authors

The authors of the manuscript “Evaluation of Spliceosome Protein SmD2 as a Potential Target for Cancer Therapy” investigate SmD2, a core spliceosome protein, as a potential cancer treatment target. Their pan-cancer analysis across 26 solid tumor types reveals that the SNRPD2 gene, which encodes SmD2, is overexpressed in most cancers and linked to poor prognosis. Using short hairpin RNA to silence SNRPD2, they assess its impact on cancer cell viability in various cell lines. Their claim is therefore that SmD2 is a selectively lethal target in cancer, suggesting its involvement in crucial cellular processes beyond mRNA splicing, and correlate its expression with responses to FDA-approved anti-tumor drugs.

Splicing proteins, both part of the core spliceosome machinery as well as alternative splicing regulators, have been widely known to be often upregulated in cancer (10.1038/s41568-022-00541-7) . The authors should mention that this is an established characteristic and cite the relevant work. Following this observation, it would be meaningful to compare the upregulation of SNRPD2 gene, on which the paper is focused on, to at least the other Sm proteins. Is it the most upregulated among those? It is not very clear why the authors focus on this particular Sm protein.

The predicted protection effect of SNRPD2 expression in LUSC and BRCA is addressed, but only briefly, and the explanation is not convincing. No difference in the hallmarks of LUAD and LUSC is shown in figure 1C left panel. Besides, the right panel does not offer a convincing explanation, but just a bioinformatic correlation that appears to be meaningless without some sort of experimental validation.

In figure 2A no indication on the number of replicates is given. More importantly, the authors do not show any data on the knockdown levels of the SNRPD2 transcript, which is a fundamental piece of information in this kind of experimental setup. The constant claim that SmD2 could be a therapeutic target clashes with the absence of information on the knockdown levels throughout and the fact that Sm proteins are essential for any cell survival. Therfore, the therapeutic window must be established first (is there a threshold of knockdown that must be hit?). The correlation shown in figure 2C is weak and not convincing.

In figure 3 the claim seems to be that the reduction of SmD2 in melanoma cells affects their viability, and a cell line generated by the authors (Mel 41) is instead not responding to the treatment. The claim is that this cell line is “presumably” not melanoma, but a control using established non-malignant cells should be done instead.

In figure 4 the data is convincing, but it would be important to see the basal levels of SNRPD2 in these cell lines compared to the cancer cell lines used in the previous figures. If the claim is that decreasing the amount of SmD2 protein can benefit cancer treatment that overexpresses this protein, a dose-response siRNA treatment should be performed, or at least be shown that succeptiplity to knock-down increases with protein/transcript levels (in figure 2A, perhaps).

In figure 6, an attempt to correlate drug-sensitivity to SNRPD2 expression is drawn. This correlation is not drawing any insight over the mechanism that the authors could potentially explore in shedding light over the specific SmD2 cancer dependency. However, it could potentially be of some use in a clinical setting to stratify patients. The AUC is however not therapeutically meaningful per se, and should be compared with the AUC of other Sm proteins at least, besides the known target of some of the drugs listed out in figure 6B as a comparison (e.g.: Sorafenib AUC with VEGFR/PDGFR).

Reviewer 2 Report

Comments and Suggestions for Authors

The authors have tried to identify a novel spliceosomal target, SmD2, for cancer therapy. The soft data with online databases with regard to this gene looks promising. In addition, the authors have tried to explore the effect of silencing the gene on cell viability in a panel of cancer cells and normal cells, as well, to conclude its effect on curbing cancer cell viability without affecting the normal cells. The english language and grammar is adequate to make this article well presentable. However, some minor directions that could be addressed are: a) Please recheck the insertions of figure numbers throughout the results so that its made sure that appropriate figure numbers are inserted at the end of each result conclusion you make. b) Since it is a anti-cancer target that you look into it would be great if you can look into other cancer parameters such as invasion, migration or metabolic responses by silencing the gene. c) Could you perform the experiments of cell viability with over-expression of the gene in cancer cells and normal cells?

Round 2

Reviewer 1 Report

Comments and Suggestions for Authors

The authors have revised the manuscript by incorporating additional experimental data and analysis. While the data presented are robust, several critical issues remain regarding the interpretation and claims made in the paper.

Firstly, the title, "Evaluation of Spliceosome Protein SmD2 as a Potential Target for Cancer Therapy", is misleading. The expression of SmD2, as shown in the newly included S1 figure, is not significantly different from other Sm proteins, and, in fact, Sm proteins have been shown to be upregulated in various tumors, similar to other splicing factors. This suggests that SmD2 does not possess unique characteristics that would make it an obvious or promising pharmacological target. As also stated by the authors Sm proteins, as "housekeeping" genes, are essential for cell survival, and their reduction could indeed be detrimental. The authors commented "The reason why we focus on this particular protein is because in another ongoing
study we have prioritized this protein for inhibitor drug discovery". However they do not mention anything about this inhibitor drug discovery pipeline. Without this important justification it is for me hard to believe on pure trust this choice, as targeting Sm expression pharmacologically presents significant challenges: the therapeutic window would likely be narrow and difficult to achieve, given their essential role in cellular function.

While the authors do not propose a specific pharmacological modality, assuming that small molecules might be considered, there are additional concerns. Sm proteins are not enzymes, making them difficult to target effectively with small molecules. Moreover, the concept of targeting Sm proteins is not novel—PRMT5 inhibitors, which impact SmD1, D3, and B/B' functionality, have already been tested clinically. If we were to assume that SmD2 could be targeted with small molecules, the authors would still face the challenge of achieving a therapeutic window, especially given the weak correlation between SmD2 expression and synthetic lethality.

The authors state, "In Figure 2a, transduction of the panel of cancer cell lines with the shRNA-expressing lentiviral vector was consistently lethal; in contrast to the empty control vector. There was thus no doubt that the cells were successfully transduced and the lethal silencing molecule was expressed." This claim, however, warrants further scrutiny. The lethality observed following shRNA transduction does not necessarily validate the specificity of the knockdown. Off-target effects are common with this methodology, and without a rescue experiment or a clear demonstration of target knockdown via qPCR, the interpretation remains questionable. At a minimum, the authors should present data quantifying the level of SmD2 knockdown to support their claim of lethality being directly attributable to this reduction. A rescue would be optimal.

Furthermore, as the title of the paper positions SmD2 as a potential therapeutic target, the authors must address the question of how much SmD2 expression is necessary for normal cells to survive, and at what threshold its reduction could selectively target cancer cells. Without this critical information, the claim that SmD2 is a therapeutic target remains speculative.

In summary, I recommend that the authors reconsider the claims regarding SmD2 as a potential therapeutic target. Removing or revising these claims would ensure the paper is more aligned with the current evidence and better suited for its intended scientific audience.

Round 3

Reviewer 1 Report

Comments and Suggestions for Authors

I thank the authors for the response to my comments and acknowledge their patience in answering to my doubts. However, most of the comments are referring to undisclosed data from other research conducted in the lab or experiments that cannot be performed due to the lack of biological material left. The novelty or knowledge introduced by this research is not sufficient for publication. SmD2 is an housekeeping gene, essential for every cell to survive, and while cancer cells (might) need higher levels of this protein to survive, this is true for hundreds of other genes. In fact, it is well known that cancers up-regulate most of the core splicing factors, and its growth depends on this. Without a clear therapeutic window or a more robust sample stratification, this gene does not represent a potential target for cancer therapy as stated in the title of the paper, at least not more than any other essential protein.